# Sewage Pollution Promotes the Invasion-Related Traits of *Impatiens glandulifera* in an Oligotrophic Habitat of the Sharr Mountain (Western Balkans)

**DOI:** 10.3390/plants10122814

**Published:** 2021-12-20

**Authors:** Milos Stanojevic, Maja Trailovic, Tijana Dubljanin, Zoran Krivošej, Miroslav Nikolic, Nina Nikolic

**Affiliations:** 1Faculty of Sciences and Mathematics, University of Priština in Kosovska Mitrovica, Lole Ribara 29, 38220 Kosovska Mitrovica, Serbia; milos.stanojevic@pr.ac.rs (M.S.); zoran.krivosej@pr.ac.rs (Z.K.); 2Institute for Multidisciplinary Research, University of Belgrade, Kneza Viseslava 1, 11030 Belgrade, Serbia; m.trailovic@imsi.bg.ac.rs (M.T.); tdubljanin@imsi.bg.ac.rs (T.D.); mnikolic@imsi.bg.ac.rs (M.N.)

**Keywords:** fecal wastewaters, Himalayan balsam, invasive alien species, soil nutrient enrichment, phenotypic plasticity, vegetation degradation

## Abstract

An annual plant, Himalayan balsam (*Impatiens glandulifera* Royle) is globally widespread and one of Europe’s top invaders. We focused on two questions: does this species indeed not invade the southern areas and does the environment affect some of its key invisibility traits. In an isolated model mountainous valley, we jointly analyzed the soil (21 parameters), the life history traits of the invader (height, stem diameter, aboveground dw), and the resident vegetation (species composition and abundances, Ellenberg indicator values), and supplemented it with local knowledge (semi-structured interviews). Uncontrolled discharge of fecal wastewaters directly into the local dense hydrological network fostered mass infestation of an atypical habitat. The phenotypic plasticity of the measured invasion-related traits was very high in the surveyed early invasion (30–50% invader cover) stages. Different microhabitat conditions consistently correlated with its growth performance. The largest individuals were restricted to the deforested riparian habitats, with extreme soil nutrient enrichment (primarily by P and K) and low-competitive, species-poor resident vegetation. We showed that ecological context can modify invasion-related traits and what could affect a further invasion process. Finally, this species is likely underreported in the wider region; public attitude and loss of traditional ecological knowledge are further management risks.

## 1. Introduction

Biological invasions are an important aspect of anthropogenic global change [1]. In general, the impact of plant species invasions on reducing native biodiversity, changing species dominance relations and community patterns, and modifying ecosystem processes has been well documented [2,3]. Around the world, about 1500 vascular plants are currently categorized as invasive aliens and this number has a tendency to escalate, in particular, in relation to global change; thus, even more action from researchers, managers, and policy makers is urged for ([4], and the references therein).

The therophyte (summer annual) plant species *Impatiens glandulifera* Royle (Himalayan balsam, fam. Balsaminaceae) is considered native to the relatively narrow range in the foothills of the western Himalayas. It was introduced to the United Kingdom 182 years ago as an ornamental and nectar producing species and very soon became established in several European countries [5]. It is currently classified as invasive in 30 countries around the world [6]. Outside of its native range, *Impatiens glandulifera* colonizes mostly nutrient-rich lowland riparian areas, using rivers as its highways [7]. There, it is often considered a problematic invader, with a potential to depress species richness and pollination in the invaded plant communities, disrupt soil biota, exacerbate soil erosion, and hamper litter decomposition [8]. The Himalayan balsam has been the focus of much research endeavors lately. In a very recent review [9], about 240 studies on this species published during the last two decades have been reported. In the EU, in particular, *Impatiens glandulifera* has become one of the 36 invasive alien plants of Union concern, which implies the obligation of the member countries to prevent unintentional introduction and commit to its eradication and management [10]. Himalayan balsam already spontaneously grows in 32 countries of Europe and is currently being increasingly reported [6,11]. Phenotypic plasticity is an important component of the invasiveness of this species [12]. 

This species is, nevertheless, extremely rarely related to the south of Europe, including the Western Balkans (Figure 1a; [6]). The assessment of species richness (including exotic and invasive species) is yet always a function of sampling efforts [13]; the lack of reports does not automatically imply the absence or an incapacity of the species to thrive in a certain environment. The area of the Western Balkans (Bosnia and Herzegovina, Serbia, Monte Negro, North Macedonia, Albania) is still characterized by an overwhelming scarcity of environmental monitoring data in general; in the area of Kosovo (this designation is strictly geographical, without any prejudice to positions on status, and in line with UNSCR 1244), where we very recently observed the presence of *I. glandulifera*, in particular, the data on systematic floristic investigations relevant for this invasion are non-existent. These circumstances implied some methodological limitations, most notably difficulties to identify suitable control sites (as discussed by [14,15]) to study the effect of Himalayan balsam on the resident vegetation. Besides, the effect of this alien invasive plant on domicile vegetation in its typical habitats of Europe (except for the southern areas) has been well investigated (reviews by [9,16]).

On the other hand, there is very limited knowledge on the effects of environment on the invasion potential of this species, though it has been pointed out that species x environment interactions can importantly modify the invasion process [17]. In this exploratory study, we analyzed the presence of the invasive alien species *Impatiens glandulifera* in a model municipality located in the mountainous area in southeast Kosovo, in a marginal, sparsely populated upper watershed, normally not conductive to this invasion. Specifically, we aimed to relate the environmental context (soil and resident vegetation) to the growth parameters (height, aboveground biomass, stem diameter) of this invader.

## 2. Results

### 2.1. The Invader Is Already Well-Established

In the area of this study (Figure 1b), Himalayan balsam is currently past the status of an ephemeral occurrence. In the study area, *Imaptiens glandulifera* was found to be well-established in the alluvial plain of the Lepenac river and along its tributaries (Figure 2). Patches of a constant minimal size (>10 m^2^) of dense colonies (estimated abundance of Himalayan balsam over 40%) cover, in a fishing net fashion, the lower elevations in the valley, from about 730 to 1180 m a.s.l. (Appendix A). The invaded area was not limited to the riverbanks; well-developed stands were found up to 900 m away from the water courses (Appendix A), either in the vicinity of settlements (in particular, it was observed to grow towards used or abandoned stables, Figure 2e) or along the regional road Prizren-Uroševac, which runs in parallel to the Lepenac flow.

Overall, besides the most common habitats with the most prominent dominance in the riparian zone, this species was also recorded to grow abundantly along field margins, wastelands, and clearings in alluvial forests, while in the settlements it was commonly found along streams and ditches, in parks and playgrounds, ruderal places, dumping sites and abandoned buildings, and, at the higher altitudes, occasionally in backyards (Figure 2 and Appendix A). 

Furthermore, it was not observed that *Impatiens glandulifera* invaded dense swards of the seminatural, occasionally mown meadows surrounding the riparian area. Namely, the two major types of grasslands adjacent to the riparian areas have so far remained free of Himalayan balsam: the drier communities dominated by *Danthonia alpina* Vest and *Koeleria pyramidata* (Lam.) P. Beauv. of the higher and mesophillous communities of the *Arrhenatheretalia elatioris* Pawl. 1928 order of the lower sections of the Lepenac flow. Very rarely a stunted individual might appear in a less vigorous, disturbed seminatural vegetation in the settlements (Figure 2c), but this appearance was always only transient, without lasting establishment. Likewise, *Impatiens glandulifera* did not invade dense stands of cereal crops in the bottom of the valley (Figure 2d). 

### 2.2. Life History Traits Are Correlated with Resident Vegetation and Soils

The visually observed differences in the size of this invasive species and the physiognomy of its chief habitats (Table 1) were quantified (Figure 3). We found an important correlation of the measured life history traits relevant for invasion with habitat characteristics (resident vegetation and soils, Figure 4). The largest individuals of this invader (Figure 2b and Figure 3; mean height, basal stem diameter, and dry weight aboveground biomass were 3.2 m, 8.7 cm, and 196 g, respectively) consistently occurred in soils with extremely elevated soil concentration of plant-available phosphorus P and potassium -K (Table 2). Compared to the non-invaded meadows surrounding the invaded river valley (Appendix A), values of plant available K in the invaded sites were 2- to 4-fold higher, while in the “A” habitats, available P was up to 14 times higher. Soil pH over 8 (“C” sites) was higher than so far reported for this invader. In general, soils of the established stands of *Impatiens glandulifera* were calcareous, rich in soil organic matter (indicated by soil organic carbon—Corg and total nitrogen—Ntot concentrations), with a neutral to slightly alkaline pH and moderate to low cation exchange capacity—CEC (Table 2). The relative position of the relevées along the Lepenac river (higher, middle, and lower sections along the Lepenac river) did not affect the floristic composition of the invaded vegetation, the *Impatiens* vigor, or the soil properties (data not shown).

Apart from the extremely elevated P and K availability, we found no other particular soil constraint for plant growth. Though the availability of calcium (Ca) and magnesium (Mg) was considered adequate, very high K levels might cause physiological deficiency of Mg for some species (K:Mg ratio < 3 in “B” habitats) or of Ca (Ca:K ratio was on average <15). Likewise, extremely high P availability could lead to physiological disruption of the uptake of some micronutrients such as zinc (Zn). We could not detect very good correlation between soil total N (Table 2) with differences in the size of the invader or the characteristics of the resident vegetation (Figure 4b); this is to be expected, as complex interplay of processes of volatilization, denitrification, leaching, and uptake by vegetation commonly affect mineral forms of N, not so much the total ones. Nevertheless, sites invaded by the largest individuals of this species supported significantly more nitrophilous resident vegetation (on the basis of Ellenberg indicator values—EIV for nutrients, Table 3) and, in particular, *Urtica dioica* (Figure 4a). Higher availability of iron (Fe) and manganese (Mn) in soils of the riparian “B” sites (Table 2) was in accordance with the higher indication of soil moisture by the resident vegetation (EIV, Table 3).

The gradients in the resident vegetation invaded by Himalayan balsam of different vigor is shown in Figure 4. The three nonmetric multidimensional scaling—NMS—axes (only the first two are presented for clarity) represent about 66% of the encountered variation in species abundance. Group comparison (multi-response permutation procedure—MRPPanalysis, Table 4) indicated that all the three categories of the resident vegetation were statistically significantly different; the strongest separation was between the “A” and the “C” sites, while the highest floristic homogeneity was found between “A” and “B” habitats. (Table 4). About 23% of all the encountered species were restricted to “B” sites and about 25% to “C”; none of the species appeared exclusively in “A” habitats. The gradual increase in the invasion potential of *I. glandulifera* (measured as plant size, Figure 3) in different habitats was accompanied by clear changes in the resident vegetation (Figure 4), and specifically with a gradual decrease in species richness and floristic heterogeneity, an increase in the abundance of nitrophilous species (Table 3), and an increase in available P in soils (Table 2). 

On the other hand, the assemblages with strongly contrasting sizes of the invader, and highly contrasting vegetation, nevertheless shared about 26% of all the species found in these two groups (“A” vs. “C”, see Appendix A). The highest species richness (Table 3) was encountered in “B” habitats, with the intermediate vigor of Himalayan balsam (Figure 3) and intermediate values of soil nutrient enrichment with P (Table 2), but with the lowest EIV for light (Table 3), indicating the highest tree cover. Indeed, the richest invaded sites (exclusively 23% of all the species recorded in the invaded plots were found exclusively in “B” habitats) were distinguished by the presence of forest herbaceous species (e.g., *Brachypodium sylvatica*, Figure 4a, but also *Melittis melissophyllum*, *Geranium phaeum*, *Carex sylvatica*, *Stellaria holostea*, *Cephalanthera rubra*, etc., see Appendix A).

In the invaded habitats with the most vigorous growth of the invader (“A”), the average relative proportion of *Urtica dioica* cover (see Figure 4a) was about double as compared to the stands with the highest species richness (“B” stands). Overall, the conspicuously lusher growth of Himalayan balsam in the riparian zones (“A” vs. “B” sites, Figure 3) coincided with: (1) higher degree of anthropogenic modifications of the environment (soil eutrophication, particularly visible in P concentrations, Table 2; and logging, reflected in higher EIV for light, Table 3); (2) decreased species richness by about 50% (Table 3); and, consequently, (3) with the increased dominance of the group of four major nitrophilous species (*U. dioica*, but also *Sambucus ebulus*, *Petasites hybridus*, *Rubus caesius* and *Arctium lappa*) by, on average, 30%.

### 2.3. Local Practices and Attitudes

Interestingly, the local land users revealed two important (mal)practices, otherwise not mentioned in the publicly available statistics, which might aid the understanding of the causes contributing to the abundant presence of the Himalayan balsam in this mountainous, rather atypical habitat. Primarily, there is no functional sewerage system for the drainage of household wastewaters in the Štrpce/Shtërpcë municipality; septic tanks for individual households are rare. Instead, household wastewaters are often directed via primitive pipelines to the nearest waterway (a tributary stream or the Lepenac river) and freely drained there, without any pre-treatment. Clean water supply is from other sources (wells or springs on higher altitudes). When asked for an explanation of this practice (with respect to environmental pollution), the common answer was that it was “not optimal and preferred, but safe, because the river has always had a power to clean itself up”. Broadly, the same attitude underlined the occasionally observed (in the context of our larger scale floristic survey in the Sirinić valley) disposal of larger waste items in the Lepenac riverbed, such as old furniture or even used cooking stoves, hoping that “the water shall carry it away”. Secondly, we have noticed a very widespread practice of taking the stable manure out to age and ferment just immediately on the banks of the river and streams (Figure 2a). This practice was allegedly established relatively recently; with the civil war in the 1990s, people felt increasingly unsafe to keep the livestock in the pastures during warm periods of the year as they used to do traditionally before the conflicts. The number of animals was reported to have drastically decreased in the last few decades and what was left was kept mostly in stables. Grazing by livestock has, hence, not been an option for *Impatiens* control in the study area in the last decades. To our surprise, nowadays this manure was subsequently rarely used as fertilizer on crop fields (which are usually located on flatter land by the waterways), but most commonly it was just left on the river/stream banks. 

Furthermore, we were not able to establish the exact time of arrival of this species in the region. Yet, we were informed by elder interviewees that, in the early 1970s, Himalayan balsam was a new, fashionable ornamental garden plant, which soon proved aggressive and difficult to control. The local, vernacular name of this plant is “*pucavac*” (“the exploding one” in Serbian), reffering to its ballistochoric seed dispersal mechanism. Moreover, allegedly, people were occasionally bringing the manually weeded herb of “*pucavac*” [pronounced *poots-awats*] on the riverbanks, together with other household waste; this practice has also fostered a faster spread of the plant in the area. Although we occasionally observed *I. glandulifera* as a weed in raspberry fields, local farmers did not perceive it as a problem. In general, we observed a very low awareness of risks related to the presence of this plant and a prominent erosion of the relevant traditional knowledge in younger generations (Table 5). The respondents could, in part, ascribe the decreasing importance of the traditional ecological knowledge to the increased possibilities for off-farm employment in the new local governmental institutions.

## 3. Discussion

### 3.1. Sewage Pollution Favors the Invasion in an Oligotrophic Habitat

We report a systematic infestation of the upper part of a medium-sized watershed in a mountainous area of Southeastern Europe (Appendix A and Figure 2), a region where this alien species has not been reported so far. On the basis of a hundred-year monitoring experience [7], it can be expected that Himalayan balsam in the study area might already be in the phase of exponential spreading. In principle, in comparison with the available reports from thoroughly investigated parts of Europe [9], we found nothing unusual about the type of vegetation Himalayan balsam invaded nor about the features of its habitat or mode of introduction and spread. The plant available concentrations of potentially harmful elements (Cd, Cr, Pb, and As) were low, below the phytotoxic levels, in a range reported for rural (not industrialized) areas [18]. We did, nevertheless, find conspicuously larger sizes (Figure 3) than so far reported for this species. We also reported pronouncedly higher availability of phosphorus(P)—and potassium (K) in the soils it occupied (Table 2) than reported so far. This species was indeed very recently assigned a maximal indication score of 5 for P availability in soils [19]. Besides, the optimal Ellenberg indicator value (EIV) for soil nutrient for this invader in Europe is 7 [20], while we found it to have the highest invasion potential in habitats with an average EIV of 8 (Table 3). 

The soil analyses did not indicate that any other analyzed parameter except extremely high P and K (Table 2) could have *per se* contributed to the differentiation of either the resident vegetation or the size of the invader. Moreover, this effect was clear only for the extremely large invasive individuals in the “C” sites (Table 2, Figure 4b). The observed differences in the life history traits of the invader were hence likely induced by other factors such as competition, and light and moisture availability (Table 3). The soils harboring this invasive species were polluted by communal wastewaters. As the uninvaded adjacent sites (Appendix A) had, by far, lower concentrations of soil P and K than the invaded ones (Table 2), and non-riparian soils in the municipality were in general oligotrophic [21], we could assume that the differences in the analyzed soil samples were chiefly caused by anthropogenic factors, primarily by sewer discharge. The potential natural vegetation of the whole surveyed area would have been alder forests; all the differences in the resident vegetation we observed were likely not the consequence of the invasion but a “black box” result of different types and severities of anthropogenic disturbance (soil pollution and logging, but also construction works such as new roads, pavements, building). Anthropogenic disturbance thus modified the abiotic environment (nutrients, light, moisture—Table 2 and Table 3) and also lowered the resilience of the resident vegetation by promoting low competitive nitrophilous species (Figure 4, Table 3). Anthropogenic disturbance, in general, and nutrient inputs, in particular, are well-known to increase the susceptibility of native vegetation to invasion [22].

In a comparable altitudinal range in the Alps, the nutrient enrichment of the riparian habitats assumed to have favored the invasion, was ascribed to runoff from the adjacent agricultural areas, increasing population density, or even the effects of the Second World War [23,24]. In our research area, however, the population density is rather low (28 inhabitants per km^2^), in a range of Mediterranean islands where *I. gladulifera* has not been reported, while in the Alps or the Central European mountains, it is two and three times higher, respectively [25]. Moreover, the share of cropland in the Štrpce/Shtërpcë municipality is low, erosion processes very prominent, and the surrounding soils shallow and low in available nutrients (see Section 4). We showed that the nutrient enrichment of the riparian zones conductive to invasion of this species was brought about by the tradition of bad hygienic practices, namely by the widespread discharge of fecal wastewaters from residential areas directly, without any pre-treatment, into the local dense hydrological network. Though extremely rare in modern Europe, a lack of proper fecal water disposal affects the livelihoods of about 1.5 billion people in developing countries worldwide [26] and of more than 50% of households in Kosovo ([27]; see also [28]). The urban pollution described in this work is hence likely widespread in the wider region.

### 3.2. Phenotypic Plasticity in Different Habitats

To our best knowledge, this work presents the first demonstration that different microhabitat conditions (characterized by the resident vegetation and concentrations of soil nutrients) consistently correlate with growth performance of *I. glandulifera* in a relatively small and isolated region. The size of the Himalayan balsam individuals (plant height, basal stem diameter, and dry aboveground biomass, Figure 3) correlated well with some easily observable characteristics of its habitats within the floodplain area of the Lepenac river and its tributaries (Table 1). The occurrence of each of these three size classes of the invader was restricted to the significantly different types of native (resident) vegetation and, to some extent, to soil properties (Figure 4, Table 2, Table 3 and Table 4). Furthermore, this phenotypic plasticity occurred in a population of a presumably single origin, since the area is physically isolated, sparsely populated, in the uppermost part of a mountainous watershed. We demonstrated that the invader’s size traits as a proxy of its invasiveness depended on the invaded habitat (characterized by resident vegetation and soils). In a larger view, we showed that ecological context can modify invasion-related traits, and thereby affect the further invasion process, as discussed by [3,17].

The variability of the measured life history traits was very high: the population means for plant height, stem diameter, and dw aboveground biomass were 2.6-, 2-, and 14-fold (respectively) higher in “A” than in “C” stands (Figure 3). The size of the Himalayan balsam individuals is a proxy for the invasion potential in terms of faster growth and increased seed production [12,29]. The effect of physical environment on these parameters is rarely studied. For instance, [30] analyzed 26 *I. glandulifera* populations from nine European regions and found that larger plants (height and dry aboveground biomass) originated from the lower geographical latitudes. The largest individuals they reported (populations from an area about 7° latitude north than our research area, altitudes below 450 m.a.s.l.) had mean values of plant height and aboveground dry weight by about 80 and 90% lower, respectively, compared to the population means in our “A” stands, and were in a range of “B” stands (Figure 3). Further, a positive correlation of plant height and soil concentrations of ammonium and organic carbon—Corg—was reported from five populations of Himalayan balsam in Ireland [29]; the tallest population had a mean height of 164 cm (less than both “A” and “B” stands, Figure 3). With respect to the general effect of habitat types on growth performance of this species, lower values were reported for aboveground dw and plant heights in alder forest compared to meadow habitat [31], while [32] found about 4-fold lower biomass in roadside compared to riparian habitats. 

In nature (non-manipulative) experiments, it is often not possible to establish causal relationships among the considered factors [31]. In fact, often only a sum effect of a set of factors can be identified, while their complex interactions remain in the domain of a “black box” [33,34]. For instance, competition of *I. glandulifera* with resident vegetation (not considered in this work) can strongly affect its biomass and height [12]. In the same line, it has remained so far ambiguous whether the presence of this species causes changes in resident vegetation or if it occurs in already degraded vegetation. It has been suggested that this species might be more of a “passenger” or a “back seat driver” (*sensu* [35]) of environmental change, a symptom rather than a direct cause of ecosystem disturbance [14,31,36,37]. Nutrient enrichment *per se* can decrease species richness of riparian communities and lead to a replacement of grasses and sedges by nitrophilous herbs such as *Urtica dioica* ([20]; Figure 4a, Table 3). *U. dioica* might be equally capable of suppressing local diversity as *I. glandulifera* [38]. In perspective, it could be expected that the main effect of the extreme increase in size of this invader would be a replacement of nitrophilous species (*U. dioica* in particular) in “C” stands, as demonstrated for the communities with very high cover of *I. glandulifera* [14,31,39].

### 3.3. Local Perception Is Not Yet Negative

On the local level, in the surveyed municipality of Štrpce/Shtërpcë, which is still a marginal rural area, we found no evidence of concern among the local population about the spreading of this species (save for being referred to as a garden nuisance, Table 5). The man-made eutrophication of riverine habitats encountered here, hardly conceivable for the Europe of the XXI century, is beyond the scope of this study (but see [40] for the plausible socio-cultural contexts of the observed phenomenon). Our findings, nevertheless, underline a drastic lack of environmental awareness in the study area. Furthermore, according to our interviewees, the species was introduced at least as early as the 1970s as a highly appreciated ornamental plant. Indeed, *Impatiens glandulifera* has been highly liked by the large part of the general public in Europe [41]. Just as with many other invasive alien species [4], it was, in general, only with the Convention on Biological Diversity in 1992, and the subsequent large scale international initiatives such as the Global Invasive Species Programs or the EU-funded projects such as ALARM and DAISIE, that the negative effects of Himalayan balsam’s invasiveness to (semi)natural ecosystems, and the accompanying necessity to control/eradicate it, gained wide attention. Contrary to a broad denigration of this species by the EU authorities, there is evidence that the perceived negative impact of *Impatiens glandulifera* might have been overestimated [16,42].

### 3.4. Himalayan Balsam Might Be Underreported in the Western Balkans

In a large part of the Western Balkans, an overall lower prioritization of environmental issues, such as invasive species, was partly brought about by the recent history of civil wars and the accompanying ravages of both nature and people. In particular, in the area of Kosovo, the problems of weak civil society and the lack of public awareness on issues of biodiversity protection, even a lack of trained staff to undertake the baseline field surveys [27,43], likely underlie why no reports on *Impatiens glandulifera* are published so far. In fact, only four records of this alien are available in the GBIF database for the whole Western Balkans [6]. Knowledge on species distributions is always based on survey efforts; even in the countries with the tradition of more than two centuries of systematic floristic investigations, the observation of the exponential spread of Himalayan balsam coincided with the beginning of the Central European Mapping Project (e.g., in Austria, [24]).

While in some southern parts of the continent this species might be genuinely rare (cf. [9]), our work indicates that Himalayan balsam might be largely underreported in the Western Balkan region. Nevertheless, the species was already anecdotally reported in Bosnia and Herzegovina in 1935 [44] and in Serbia in 1995 [45]. In the wider surrounding of our research area, this invader appeared on the GBIF map only after 2014 (the report of [46] in North Macedonia). Actually, [46] recorded this species also in the Sharr Mountain (only about 10 km aerial distance from our research area) and specified that it was found “near streams that spring from the Sharr Mountains”. To our best knowledge, such environments (natural and management context) are rather typical for rural regions in the mountainous areas of the neighboring countries. This arouses a reasonable doubt on whether Himalayan balsam is already present in a wider region of the Western Balkans, including other areas of Serbia, Montenegro, Albania, and North Macedonia.

### 3.5. Implicit Risks for the Region

Though *Impatiens glandulifera* in Europe mostly occurs in lowland habitats [9], it was anecdotally observed in all the mountainous/hilly areas of West and Central Europe, occasionally even at altitudes higher than 1200 m (cf. [23]). In this work, we found it to be well-established at elevations below 1200 m, but we did not investigate higher altitudes. It would take just 300 m of altitudinal difference for this plant to “climb up” and reach the Sharr Mountain National Park, which is located immediately above our research locality, at elevations from 1500 to over 2500 m a.s.l. This nature reserve is one of the key biodiversity hotspots in the Western Balkans and the largest refugium of glacial flora on the continent, harboring about 18% of the European species. Similar concern has already been raised for the Tatra National Park in Poland, in relation to global warming in particular [15]. Furthermore, the fluvial dynamics of the investigated part of the Lepenac watershed are such that the erosion is, by far, more prominent than the accumulation processes; about 74,000 km^3^ of sediments (i.e., about 72% of the sediments generated in the watershed) are washed away from the investigated municipality every year [47], which very likely fosters considerable downstream transport of the Himalayan balsam propagules. Finally, all the well-known drawbacks of a weak civil society, which include the current lack of public awareness on risks related to invasive species, as well as inadequate relevant legislation framework, are likely to make the management and control of Himalayan balsam in the region rather difficult. 

## 4. Material and Methods

### 4.1. Study Area

The study was conducted in the alluvial plain of the river Lepenac, in the so-called Sirinić tectonic depression (municipality of Štrpce/Shtërpcë), within the Sharr Mountain complex in the southeast of Kosovo (Figure 1b). This valley is spatially rather isolated, surrounded by very high mountain massifs of about 2500 m a.s.l. The survey area stretched along about 19 km of the Lepenac flow, from the village of Sevce (at about 1200 m.a.s.l.; 42°12′47.86″ N, 20°56′07.26″ E, where the river Lepenac is formed), to the village of Drajkovce (altitude about 730 m.a.s.l; 42°15′26.51″ N, 21°04′57.23″ E), and also included the area along the ten major tributaries of the Lepenac up to the elevation of 1200 m a.s.l. All the surveyed tributaries passed through or beside human settlements. The Lepenac is a typical mountainous river, formed in our research area; its basin (total area about 770 km^2^, a third of it belongs to the municipality of Štrpce/Shtërpcë and has an average slope of 50 m per km) drains to the Aegean Sea. In the municipality, the total length of the Lepenac and its numerous tributaries is about 360 km (about 1.5 km of waterways per km^2^, mostly streams of about 2–5 km long); it leaves Štrpce/Shtërpcë with the average discharge of about 5.6 m^3^ s^−1^ [48]. The flow and the flooding regime of the Lepenac are completely free and unregulated in the study area.

The municipality of Štrpce/Shtërpcë is a marginal rural area encompassing 16 settlements on about 247 km^2^ and average density of only about 28 inhabitants per km^2^; 11 settlements are on the elevations from 600 to 1200 m.a.s.l. The population (ethnic Serbs and Albanians) is, in the last decade, facing an annual decrease of about 0.5% (Kosovo Agency of Statistics, https://askdata.rks-gov.net, accessed on 15 October 2021). The local people have traditionally subsided chiefly on animal husbandry (sheep and cattle); cropland occupies only about 10% of the total land in the municipality. The relief is pronouncedly dissected, with a third of the area on slopes from 20 to 30°, and the average slope of hydrological network about 20‰. The soils on higher altitudes, above the Lepenac valley, are mostly shallow, skeletal, and oligotrophic, poor in plant available phosphorus (P)—below 10 mg kg^−1^ AL-extractable P fraction [21]. The climate is a modified continental with a prominent sub-Mediterranean influence. The average annual precipitation (measured at the station Brezovica, at 915 m a.s.l.) is 974 mm and the average annual temperature 7.9 °C (on the basis of the data available for the previous climate normal period, 1960–1990). The potential natural vegetation of the study area would be, to the largest extent, thermophilous oak forests of the association *Quercetum frainetto–cerridis* Rudski (1940) 1949 *sensu lato*, i.e., more precisely, of a geographic variety with the pronounced sub-Mediterranean character, locally termed *Quercetum frainetto-cerridis scardicum* Krasn.1968 nom. inval. [49]. The relatively narrow riparian zone of the Lepenac river is dominated by alluvial non-swamp forests of black alder, *Alnus glutinosa*, with the presence of *Sambucus nigra*; [50]). The municipality of Štrpce has, in the 20th century, been profoundly modified by anthropogenic disturbance, primarily by severe deforestation for agriculture. In the flatter parts of the landscape, the forest cover is, nowadays, reduced to the linear fragments of alder stands along the waterways. The lowest altitudes (relatively flat portions of the valley) are used as mown meadows and crop fields, while pastures, degraded grasslands, and a successional matrix of fallow land occupy steeper slopes and higher elevations (satellite image available at https://www.google.com/maps/@42.2359,21.02695,12493m/data=!3m1!1e3, accessed on 17 September 2021). The massive reconstruction of the infrastructure in the municipality (roads—particularly the regional road Prizren-Uroševac, pavements, buildings, parks, etc.) was initiated about a decade ago, and the sand dug from the Lepenac riverbed has been used for the works. Up to now, *Impatiens glandulifera* is not listed as an alien species in the territory of Kosovo (Kosovo Environmental Protection Agency 2020, available at https://www.ammk-rks.net, accessed on 23 September 2021); it is considered “sporadically invasive” in Serbia, though the relevant legislative framework for invasive species management is still lacking.

### 4.2. Vegetation Survey

For the purpose of studying the effect of environment (resident vegetation, soils) on the growth performance of *I. glandulifera,* we selected sampling sites where the estimated cover of this invader was in a range 30–50%. Thereby, we have excluded the possibility of a strong effect of the invader on species composition of resident vegetation, as these effects were reported at its much higher cover (usually over 80%, e.g., [15,37]). Furthermore, we surveyed the vegetation in 16 m^2^ relevées; they were not always of quadratic shape, as this invader is known to have pronouncedly patchy distribution [51], and we aimed to select the sampling sites with as much as possible dispersed (not clumped) distribution of the Himalayan balsam individuals. Samples of each of the three classes included upper, middle, and lower sections of the Lepenac flow. The distance among the relevées was >100 m.

Vegetation survey was undertaken in August 2019 and 2020 when this species was in full flowering. Sampling was carried out according to the flexible systematic model [52], a form of stratified sampling where samples were allocated on the basis of the very prominent observed difference in the height of the *I. glandulifera* individuals in the period of full flowering. Samples were stratified in three classes based on the visual estimation of the size of the individuals in the dominant layer of the invader: very large (“A”), intermediate (“B”), and small (“C”); each stratum contained 15 samples. Percentage cover of each species in each relevée was estimated with the aid of cover diagrams, and a total of 45 vegetation samples were analyzed. Life history traits (plant height, stem diameter at 10 cm above ground, and plant aboveground dry biomass) were based on 5 randomly selected individuals in each relevée (totaling 225 measured individuals). The internet associated W3 TROPICOS nomenclatural database of the Missouri Botanical Garden, and the associated authority files (http://www.tropicos.org/Home.aspx, accessed on 1 October 2021), were used as a reference.

The overall presence of *I. glandulifera* in the research area was recorded during series of systematic transect walks parallel to the flow of the Lepenac river, and 10 of its major tributaries (streams), up to the elevation of 1200 m.a.s.l. Occurrences of “significant” stands were recorded with GPS coordinates. The threshold for significant was an area of at least 10 m^2^ with the estimated abundance of Himalayan balsam ≥40%, or the presence of at least 50 individuals. Sporadic occurrences of the species were not considered. 

### 4.3. Soil Survey and Analyses

Soil sampling was conducted in August, during the vegetation surveys. In each relevée, a composite sample (consisting of 3 randomly chosen subsamples) was obtained by soil core at 0–30 cm depth (rooting zone), totaling 45 samples. In addition, 15 soil samples were taken from the surrounding vegetation of (semi)natural grasslands, immediately outside the flooding zone of the Lepenac river, along its course in the research area. After drying and sieving through a 2-mm mesh screen, the samples were subjected to analyses. Briefly, the soil pH was measured in soil:water extract (1:2.5), and the concentrations of total carbon (C), nitrogen (N), and sulfur (S) were determined by the CNS analyzer (Vario Micro Cube, Elementar Analysensysteme GmbH, Hanau, Germany). The total CaCO_3_ was determined by the Scheibler calcimeter, and organic carbon (Corg) was calculated by subtracting total CaCO_3_ from the total C. For determination of plant-available fraction of different elements, the following extractants were used: Olsen for P; ammonium acetate for Ca, Mg, K, and Na; DTPA-sorbitol for B; and DTPA-TEA solution for Fe, Cu, Zn, Mn, Cd, Ni, Pb, Cr, and As. The elemental concentrations (with the exception of P) were determined by inductively coupled plasma optical emission spectroscopy (ICP-OES, SpectroGenesis EOP II, Spectro Analytical Instruments GmbH, Kleve, Germany), whereas the P concentrations were determined by the colorimetric molybdenum blue method. Cation exchange capacity (CEC) was calculated from the sum of the extracted base cations. 

### 4.4. Local Knowledge Survey

The local knowledge on the Himalayan balsam was assessed by open-ended, semi-structured interviews (a group of Participatory Rural Appraisal tools described by [53]). The selection of interviewees was opportunistic and included a total of 41 residents of the Serb ethnic group (from 7 out of 11 villages located below 1200 m.a.s.l. in the Sirinić valley) who were willing to participate. The respondents were divided in two age groups: younger (16–30 years) and elder (30–60 years). All of them came from farmers’ families who used to have free-grazed livestock before the civil war in the 1990s. Interviews were conducted in August of the two consecutive years (2019 and 2020), when *Impatiens glandulifera* was in full flowering.

### 4.5. Statistical Analyses

Unconstrained ordination of the invaded vegetation samples was performed by nonmetric multidimensional scaling (NMS) using relative Sørensen distance and orthogonal principal axes; the final stress for a 3-d solution was 16.43. Relative Sørensen distance has the same effect as relativization by sample unit totals and tends to emphasize shifts in relative proportions of species when total sample covers differ considerably. Comparison of the resident vegetation in the three classes of *I. glandulifera* growth vigor was conducted by nonparametric multi-response permutation procedure (MRPP). This analysis provides a T statistic, whose more negative values indicate stronger separation among pre-defined groups of samples, and chance-corrected within group agreement (A), which describes the homogeneity within the groups as compared to the random expectation. Multivariate analyses were performed in PC-ORD 7.08 software (MjM Software Design, Gleneden Beach, OR, USA). Ellenberg indictor values (EIV, Ellenberg and Leuschner, 2010) for each relevée were calculated as averages weighted by the species cover value. *Impatiens glandulifera* and rare species with estimated cover less than 1% were excluded from the analyses of the resident vegetation. Univariate ANOVA followed by Tukey’s Honestly significant test, using a conventional level of significance of 5% (α = 0.05) (STATISTICA 6 software, StatSoft Inc., Tulsa, OK, USA), was used for comparison of means.

## 5. Conclusions

This is the first report of a severe infestation of a mountainous area in the Western Balkans by *Impatiens glandulifera*. Soil eutrophication through direct discharge of untreated household wastewaters to the local dense hydrological network, and an overall mismanagement of waste favored its establishment, both in urban and severely disturbed riparian areas. This invasive alien exhibited high phenotypic plasticity of the three life history parameters in a rather small and isolated river valley. In this explorative study, we demonstrated that *I. glandulifera* size traits, as a proxy of its future invasiveness, depended on the patchy microhabitats (characterized by resident vegetation and soils) created by different anthropogenic disturbances. The extremely large individuals (mean values of 3.2 m, 8.7 cm, and 196 g for plant height, stem diameter, and aboveground dw per plant, respectively) were consistently restricted to the deforested riparian habitats with extreme soil nutrient enrichment (primarily by P and K) and low-competitive, species-poor resident vegetation. Both the natural (source area of the river which drains to the Aegean Sea, with strong erosion processes) and societal (loss of traditional ecological knowledge and pronouncedly low public awareness on environmental issues) context imply a high risk for farther spread and difficult control of this species in the wider region.

## Figures and Tables

**Figure 1 plants-10-02814-f001:**
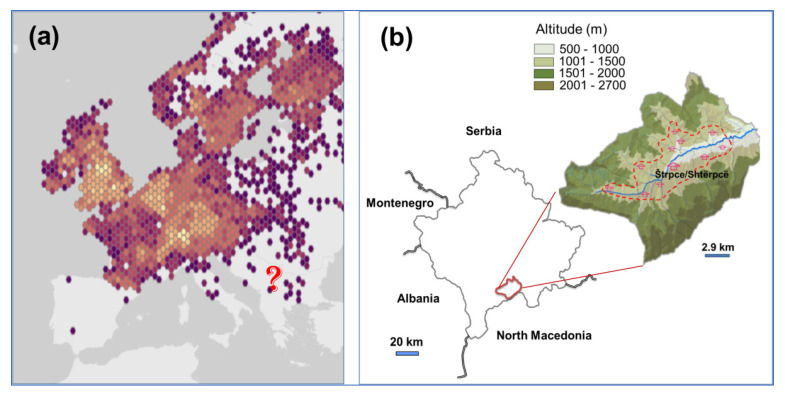
(**a**) Current distribution range of *Impatiens glandulifera* in Europe, based on GBIF presence data (based on [6]); color warmth relates to point density. (**b**) Municipality of Štrpce/Shtërpcë, in a valley of the Sharr Mountain complex, southeast Kosovo. House drawings represent settlements. Dashed line delineates the surveyed area.

**Figure 2 plants-10-02814-f002:**
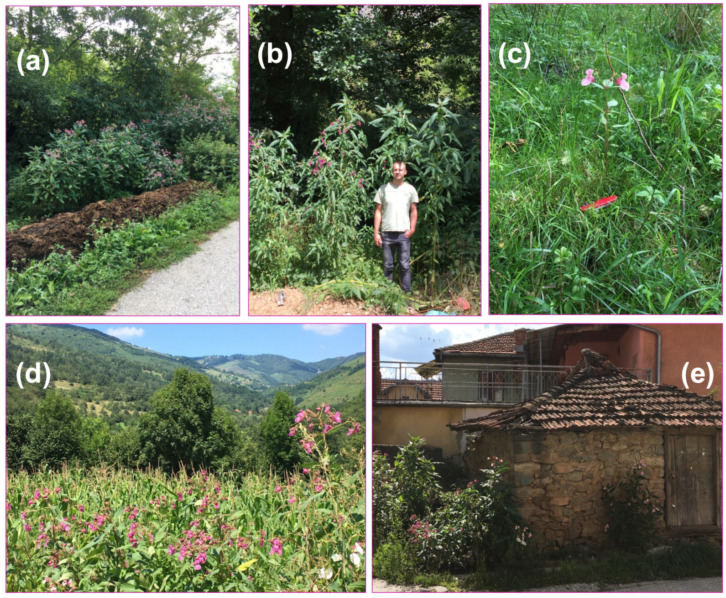
The invasion of *Impaties glandulifera* in the study area. (**a**) The practice of disposing of farmyard manure (and other organic waste) beside the waterways strongly increased nutrient availability and promotes the invasion process. (**b**) Consequently, very lush and vigorous plants (here, height >300 cm; the first author posing is 1.89 m tall!) were encountered in nutrient-enriched areas. (**c**) Rarely, stunted individuals appeared in the surrounding seminatural vegetation, but did not persist to the next generation. (**d**) Outside the riparian zone, the species appeared on ruderal sites but did not invade the adjacent dense vegetation; here, maize fields. (**e**) It is a well-established constituent of the urban flora in the municipality of Štrpce/Shtërpcë; here, concentrated around an abandoned stable.

**Figure 3 plants-10-02814-f003:**
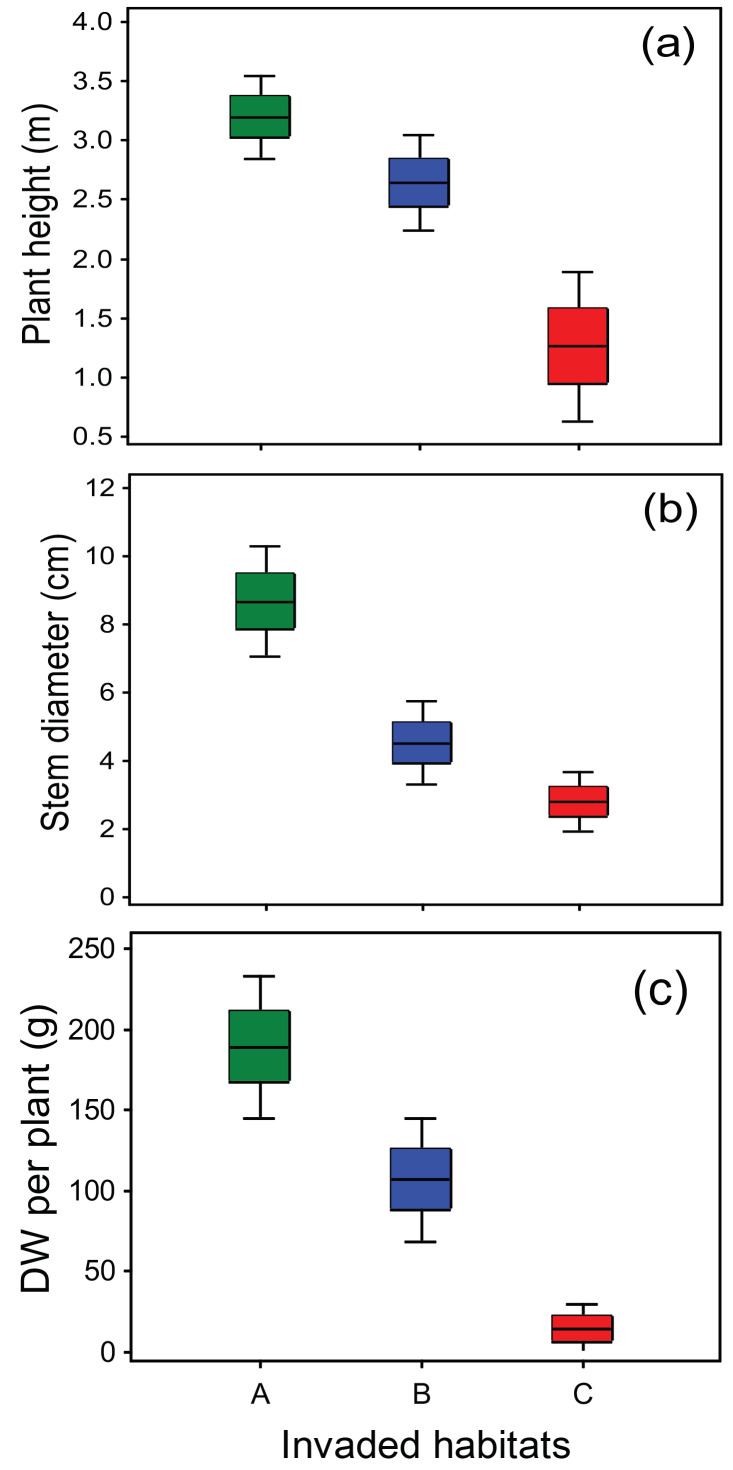
Frequency distribution (box plots, scaled in standard deviations) of life history traits of Himalayan balsam in the research area. (**a**) Plant height; (**b**) basal stem diameter; (**c**) dry aboveground biomass. The average estimated cover of this invasive species in the surveyed plots ranged from 30 to 50%. Sites are coded as in Table 1. Measurements are based on 75 individuals in each of the three types of invaded habitats. For each measured sized parameter, the differences among habitat types were statistically significant (*p* < 0.05, Tukey’s test).

**Figure 4 plants-10-02814-f004:**
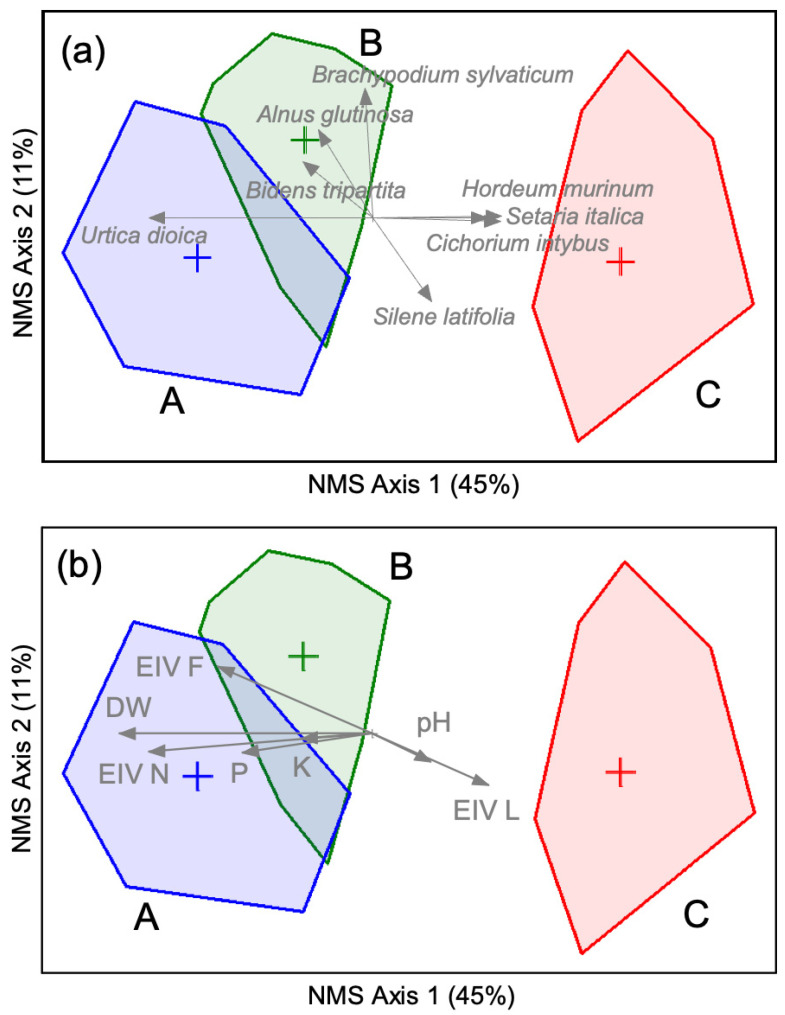
Gradients in the vegetation invaded by *Impatiens glandulifera* (NMS ordination). Invaded habitats/sites (“A”, “B”, and “C”) are defined as in Table 1. Data matrix: 45 vegetation samples, 88 species, relative Sørensen distance. The angles and lengths of the radiating lines indicate the direction and strength of relationships of the species abundance with the ordination scores. The share of the total variance attributed to each axis is parenthesized; group centroids and convex hulls are presented. (**a**) Plant species having absolute rank correlation with NMS ordination scores over 30% are overlaid. Botanical references of species are given in Appendix A. (**b**) Selected habitat characteristics correlated with NMS axes by >20% are passively projected on the ordination of species. Soil characteristic: pH, plant available concentrations of P and K; EIV F, EIV L, EIV N: Ellenberg indicator values for moisture, light, and nutrients, respectively; DW: dry aboveground biomass of *Impatiens glandulifera*.

**Table 1 plants-10-02814-t001:** Individuals of *I. glandulifera* of distinctive size and visual appearance commonly invade distinctive disturbed habitats in the valley of the Lepenac river.

Invader Physiognomy	Typical Invaded Sites	Code
Prominently lush, coarse, average height >2.5 m	Highly disturbed riparian but also moist non-riparian areas, visible deposition of organic waste (garbage, manure), sparse tree canopy	A
Intermediate	Riparian area with denser forest cover and no obvious garbage deposition (lower anthropogenic influence)	B
Slender, average height <1.5 m	Open ruderal habitats, roadsides, deposits of sand and other construction waste, often also household waste; vicinity of waterways, but not flooded	C

The codes “A”, “B”, and “C” thereafter refer to both invader’s size and site types.

**Table 2 plants-10-02814-t002:** Soil properties of the invaded habitats with different vigor of *Impatiens glandulifera*.

Soil Parameter	Site Code		
	A	B	C
pH	7.6 ± 0.3 a	7.2 ± 0.4 a	8.1 ± 0.2 b
CEC (cmolc kg^−1^)	22 ± 5 b	15 ± 3 a	24 ± 4 b
CaCO_3_ (%)	4 ± 2 b	2.0 ± 1 a	8 ± 2 c
Corg (%)	3.4 ± 0.9 b	3.6 ±1.2 b	2.8 ± 0.6 a
Ntot (%)	0.41 ± 0.05 b	0.46 ± 0.04 b	0.31 ± 0.05 a
Stot (%)	0.07 ± 0.03 b	0.02 ± 0.02 a	0.02 ± 0.01 a
Available P (mg kg^−1^)	146 ± 49 c	28 ± 7 b	18 ± 5 a
Available K (mg kg^−1^)	863 ± 229 b	421 ± 155 a	418 ± 174 a
Available Ca (mg kg^−1^)	3363 ± 1181 a	2379 ± 635 a	4372 ± 709 a
Available Mg (mg kg^−1^)	270 ± 82 a	227 ± 72 a	145 ± 40 a
Available B (mg kg^−1^)	0.16 ± 0.08 a	0.06 ± 0.02 a	0.11 ± 0.04 a
Available Fe (mg kg^−1^)	53 ±18 b	64 ± 9 c	34 ± 9 a
Available Mn (mg kg^−1^)	13 ± 3 a	20 ± 1 b	11.1 ± 0.7 a
Available Cu (mg kg^−1^)	4 ± 1 b	2.7 ± 0.5 a	2.0 ± 0.4 a
Available Zn (mg kg^−1^)	7 ± 2 b	1.9 ± 0.5 a	3.6 ± 0.8 a
Available Mo (mg kg^−1^)	0.026 ± 0.002 a	0.02 ± 0.05 a	0.0289 ± 0.0008 a
Available Ni (mg kg^−1^)	1.2 ± 0.6 a	4 ± 1 b	0.7 ± 0.2 a
Available Cr (mg kg^−1^)	0.03 ± 0.01 a	0.04 ± 0.01 a	0.03 ± 0.01 a
Available Cd (mg kg^−1^)	0.03 ± 0.01 a	0.04 ± 0.01 a	0.03 ± 0.01 a
Available Pb (mg kg^−1^)	3.6 ± 0.9 a	2.7 ± 0.7 a	3.8 ± 0.9 a
Available As (mg kg^−1^)	0.02 ± 0.02 a	0.05 ± 0.03 a	0.04 ± 0.01 a

Habitat codes and vigor of the invader were defined in Table 1; see also Figure 3. The estimated cover of the invader was 30–50%. Mean values ± s.d. followed by the same letter in a row are not different (*p* < 0.05, Tukey’s test).

**Table 3 plants-10-02814-t003:** Selected characteristics of the seminatural vegetation of the invaded habitats with different life history traits of *Impatiens glandulifera*.

Resident Vegetation	Site Code		
	A	B	C
Average species number per sample	7.5 ± 1.5 a	14.3 ± 3.2 b	7.0 ± 1.8 a
Total number of species ^a^	39	53	46
Weighted mean distance (relative Sørensen)	0.66	0.76	0.82
Ellenberg indicator value for light	7.0 ± 0.5 b	5.7 ± 0.2 a	7.9 ± 0.6 c
Ellenberg indicator value for moisture	6.8 ± 0.5 b	7.9 ± 0.3 c	5.4 ± 0.4 a
Ellenberg indicator value for N	8.1 ± 0.5 c	6.7 ± 0.4 b	5.5 ± 0.4 a

^a^ recorded in 15 quadrates per invader’s size class. Resident vegetation and life history traits of the invader were defined in Table 1 and Figure 3. The vegetation was sampled in 45 relevées of 16 m^2^ area; the average estimated cover of the invasive species ranged from 30 to 50%. Mean values ± s.d. followed by the same letter in a row are not different (*p* < 0.05, Tukey’s test).

**Table 4 plants-10-02814-t004:** MRPP analysis of the resident vegetation types hosting different size classes of Himalayan balsam.

Comparison of the Invaded Resident Vegetation	Test Statistics		
	T	A	*p*
“A” vs. “C”	−16.1	0.12	0.00000000
“B” vs. “C”	−13.6	0.11	0.00000000
“A” vs. “B”	−7.6	0.05	0.00000024

Resident vegetation was defined in Table 1, see also Figure 4. Data matrix: 45 samples, 88 species, relative Sørensen distance. Overall comparison: T = −18.7, A = 0.12, *p* < 10^−8^.

**Table 5 plants-10-02814-t005:** Local knowledge of the exotic *Impatiens glandulifera* in the Sirinić valley, southeast Kosovo.

Local Attitudes	Respondent Age Group	
	Young (16–30 y)	Elder (30–60 y)
	% of Responses	
Awareness		
Know it is present	44	86
Know it is introduced/non-native	0	6
Familiar with its vernacular name	0	15
Perception		
Indifferent to its presence	97	74
Amenity of riparian landscape	18	5
Nuisance weed in gardens	3	23

The selected results from the 41 interviewees are summarized. Multiple perceptions were allowed.

## Data Availability

All the relevant data are presented in Tables, Figures, and Appendix A of this paper.

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
