# Peer review of "Sewage Pollution Promotes the Invasion-Related Traits of Impatiens glandulifera in an Oligotrophic Habitat of the Sharr Mountain (Western Balkans)"

_plants, 2021, doi:10.3390/plants10122814_

Round 1

Reviewer 1 Report

The manuscript is well set up. The experimental design could be clearer and better described

it is preferable to insert after the introduction the paragraph materials and methods, then results, conclusions and discussions

check the scale in figure 1b as 2,9Km or 20 Km

check  vegetation associations write in italics 

In Italy cfr reference: Celesti-Grapow L. & al. (2009) The inventory of the non-native flora of Italy. Plant Biosystems 143(2):386-430.

Author Response

Dear Reviewer, thanks for your patience and constructive suggestions. We have carefully addressed all the points you made as following:

  1. The manuscript is well set up. The experimental design could be clearer and better described.

RESPONSE 1: Thank you. As for the experimental design, please do note that this is essentially an exploratory study (l. 78-79 in the first submission); the reasons why the study is exploratory were also explained (l. 66-75 in the first submission). We (and the other two reviewers) are absolutely convinced that a) objectives of our study are important; b) the design is suited to address the main research question (l. 78-83 in the first MS submission); and c) interpretation and discussion are well-supported by the study design.

Of course, it could have always been better. For instance, it would have been nice if we could also include some interviewees of the Albanian ethnic group living in the valley, but it was not possible (total lack of supportive infrastructure). It would also be nice if we could have some control sites (e.g. no Impatiens  ± pollution, or +Impatiens - pollution) or if we could even have any vegetation- land cover-, or any data at all about this invader from 5, 10 or 40 years ago. But this data does not exist, and we had to live with what was available. Thanks for the understanding.

  1. it is preferable to insert after the introduction the paragraph materials and methods, then results, conclusions and discussions

RESPONSE 2: We do agree, of course, but this is the requested template of this Journal. We can do nothing about it.

  1. check the scale in figure 1b as 2,9Km or 20 Km

RESPONSE 3: Thanks, we did check, and the both scales in Figure 1 are ok. We are aware that it should be more logical to put a length unit of 3 km instead of 2.9. We beg your understanding that in the current complicated relations between Serbia and Kosovo, we were not in the best position to choose the maps and ask for some specific ones. We did try, but at the end we had to relay on what was available. So our maps are absolutely accurate, but we know that, theoretically, they could also have looked nicer.

  1. check  vegetation associations write in italics 

RESPONSE 4: Done.

  1. In Italy cfr reference: Celesti-Grapow L. & al. (2009) The inventory of the non-native flora of Italy. Plant Biosystems 143(2):386-430.

RESPONSE 5: Yes, we are aware of this nice publication. It shows that “our” plant is invasive in Italy; we have re-checked the GBIF database for the Himalayan Balsam in Italy and saw that perhaps one observation (from San Marino) might indeed be from Mediterranean/coastal area. Since our work is exclusively focused on the Western Balkans, we did not include this nice paper on Italy. We did, however, accept your suggestion and deleted a reference to the absence of this species in Mediterranean areas (line 400 in the first submission).

Reviewer 2 Report

Only minor comments:

  • The titles of Tables should not contain concrete details, such as statistics, decoding of A,B,C areas, data matrix, details of vegetation, amount of interviewers (Tables 1-5). All these data should be places below the Tables as foot notes.
  • Abbreviations used in the text should be decipher in Results and Discussion sections but not only in Material and Methods (EIF, CEC, Stot, Corg, NMS, etc). Otherwise it is difficult to understand the data as according to Plants’ ‘Material and Methods’ section is placed at the end of the manuscript.
  • Reference list should be revised according to Plants’ guidelines. As an example: the article citation should be as follows:
  • Tylianakis, J.M.; Didham, R.K.; Bascompte, J.; Wardle, D.A. Global change and species interactions in terrestrial 615 ecosystems. Lett. 2008, 11, 1-13. https://doi.org/10.1111/j.1461-0248.2008.01250.
  • Check ‘CaCO3’- It should be ‘CaCO3’ everywhere.
  • Line 106 a misprint: ‘besidies' should be changed to 'besides'

Author Response

Dear Reviewer,

Thanks for thoroughly reading and improving our MS. We have carefully addressed all the points that you raised, as following:

  1. The titles of Tables should not contain concrete details, such as statistics, decoding of A,B,C areas, data matrix, details of vegetation, amount of interviewers (Tables 1-5). All these data should be places below the Tables as foot notes.

RESPONSE 1: done.

  1. Abbreviations used in the text should be decipher in Results and Discussion sections but not only in Material and Methods (EIF, CEC, Stot, Corg, NMS, etc). Otherwise it is difficult to understand the data as according to Plants’ ‘Material and Methods’ section is placed at the end of the manuscript.

RESPONSE 2: Done. We spelled out each of the abbreviations at the first mention both in Results and in Discussion Sections.

  1. Reference list should be revised according to Plants’ guidelines. As an example: the article citation should be as follows:

Tylianakis, J.M.; Didham, R.K.; Bascompte, J.; Wardle, D.A. Global change and species interactions in terrestrial 615 ecosystems. Lett. 2008, 11, 1-13. https://doi.org/10.1111/j.1461-0248.2008.01250.

RESPONSE 3: Thanks, we did revise all the references according to the requested Plants’ template.

  1. Check ‘CaCO3’- It should be ‘CaCO3’ everywhere.

RESPONSE 4: done.

  1. Line 106 a misprint: ‘besidies' should be changed to 'besides'

RESPONSE 5: done.

Reviewer 3 Report

The paper investigates the effect of the sewage pollution on the invasive potential of Impatiens glandulifera in a mountainous area of the Western Balkans and the effect of soil parameters and vegetation on the physiognomy of the species in different ecological contexts.

The paper is well written, the aims well explained, as well as the study area.

Results are well presented, tables and figures are clear and appropriate.

Discussion is carefully commented. The aim of the research is achieved.

The methodology is detailed

The conclusions provides useful informations for the control of this invasive species.

Only some notes:

Delete lines from 564 to 570

Line 467: Quercetum frainetto-cerridis scardicum should be in italic. Notice: this is an invalidated syntaxon

Line 469: Alnetum glutinosae fluviatile B.Jov (195391985 is not an accepted name

Author Response

Dear Reviewer, thank you for carefully reading and improving our MS, and for well-intended constructive suggestions. We have addressed all the points you raised as following:

  1. Delete lines from 564 to 570

RESPONSE 1: Done, thank you.

  1. Line 467: Quercetum frainetto-cerridis scardicum should be in italic. Notice: this is an invalidated syntaxon

RESPONSE 2: Thanks, we did italicize all the vegetation associations in the revised MS. Yes, we are aware that this taxon is invalidated. Yet, it is still broadly used in the literature from the Balkans. We have revised the considered sentence as following:

“The potential natural vegetation of the study area would be, to the largest extent, thermophilous oak forests of the association Quercetum frainetto–cerridis Rudski(1940)1949 sensu lato, i.e., more precisely, of a geographic variety with the pronounced submediterranean character locally termed Quercetum frainetto-cerridis scardicum nom. inval.Krasn. 1968 [49].” We have also added a new reference [49 in the revised MS ]where this syntaxon is in details described.

  1. Line 469: Alnetum glutinosae fluviatile B.Jov (195391985 is not an accepted name

RESPONSE 3: We got aware of it, thanks. Please note that the last comprehensive country-level syntaxa overview for Serbia was done about four decades ago, with no revisions/updates meanwhile.  We checked, our alder forests are compatible with the description of suballiance Alnion glutioso-incanae Oberd. 1953.

Anyway, we mentioned these two syntaxa (based on the available literature) exclusively to refer to the former/potential vegetation of the research area (in M+M Section). At the time of our survey, when most of the “original” forest vegetation was destroyed long while ago, we were not able to conduct the phytocoenological analysis ourselves, but had to relay on secondary literature data, from the times when large areas of forest still existed in our research area. Since, however, none of us is really a syntaxonomist, and the work is not on syntaxonomy, we modified the sentence in question as following:

“The relatively narrow riparian zone of the Lepenac river is dominated by alluvial non-swamp forests of black alder, Alnus glutinosa, with the presence of Sambucus nigra; [described by 50].”